# Can women's 3E index impede short birth interval? evidence from Bangladesh Demographic and Health Survey, 2017–18

**Fatima Tuz-Zahura** [ORCID] *, **Kanchan Kumar Sen** [ORCID], **Shahnaz Nilima, Wasimul Bari**

Department of Statistics, University of Dhaka, Dhaka, Bangladesh

* zahura.fatima@du.ac.bd

**Data Availability Statement:** The Bangladesh Demographic and Health Survey (BDHS), 2017-18 data used in this study are available in the DHS website. To get this data set without paying any

## Abstract

### Background

Women's empowerment, education, and economic status are jointly introduced as women's 3E. A number of studies found the significant association of these three variables with maternal health outcomes, but no studies, to the best of knowledge, have been found to justify the joint influence of women's 3E on the birth interval. As several studies have revealed that the short birth interval increases the risk of adverse maternal, perinatal, and infant outcomes and it is also responsible for increasing the country's population size, more research is needed on the birth interval. Therefore, the present study aimed to investigate the influence of women's 3E on the short birth interval after controlling the other selected covariates.

### Methods

Data from the Bangladesh Demographic and Health Survey (BDHS), 2017–18 have been used to serve the purpose of the study. To measure the birth interval, at least two live births for non-pregnant mothers and at least one live birth for currently pregnant mothers born in the 5 years before the survey were included in the study. The Chi-Square test was applied to know the unadjusted association of the selected covariates including women's 3E with the short birth interval. In order to find out the adjusted association of women's 3E with the short birth interval, sequential binary logistic regression models have been used.

### Results

The study found that about 23% of births in Bangladesh were born in a short birth interval. The likelihood of subsequent births of women decreases with an increase in the score of women's 3E before or after controlling the characteristics of women, child, and households. The results of the final model show that mothers with the coverage of 50% - 75%, 75% - 100%, and full coverage (100%) in 3E have a 23%, 41%, and 42% lower odds of having short birth interval compared to mothers with coverage of below 50% in 3E, respectively.

charge one has to register using the link https://dhsprogram.com/data/dataset/Bangladesh_Standard DHS_2017.cfm?flag=1.

**Funding:** The author(s) received no specific funding for this work.

**Competing interests:** The authors have declared that no competing interests exist.

## Conclusion and recommendation

Bangladesh still lags behind in meeting the minimum requirements for inter-birth intervals set by the World Health Organization. The study has shown that the 3E in women can contribute in prolonging the duration of subsequent births in Bangladesh. Policy-making interventions are needed to raise awareness among uneducated, under-empowered and economically poor reproductive women through family planning and fertility control programs so that the country can achieve the desired fertility rate.

## Introduction

Birth interval, defined as the length of two consecutive live births, plays an important role in changing the structure of the population of a country. Moreover, health promotion policies for women and children are mainly governed by the birth spacing. The World Health Organization (WHO) technical consultation on birth spacing has recommended a minimum of 33 months between two consecutive live births or 24 months for birth to conception of a pregnancy to reduce the risk of adverse maternal, perinatal, and infant health outcomes [1–4]. A short birth interval, defined as the interval below the WHO recommendation, is responsible for an increased risk of anemia, malnutrition, toxemia, uterine rupture and self-induced abortion, gestational diabetes, placental bleeding, and even maternal mortality [2, 5–11]. It has an adverse impact on neonatal and child health such as preterm births, low birth weights, infant mortality, and it also affects their standard physical and intellectual growth [8, 12–20]. Several studies have shown that babies born at least two years apart are more likely to survive and have a lower risk of premature birth, low birth weight, and malnutrition [18, 21]. In addition, a study conducted for developing countries revealed that babies born at an optimum birth interval of three to five years after their previous siblings are about 2.5 times likely to survive compared to babies born at below two-year of intervals [3]. Studies found that an estimated 2.6 million stillbirths occur each year, and most stillbirths during pregnancy can be prevented by optimizing the birth interval [22–24]. Again, an optimum interval can also create positive health outcomes for mothers, where mothers can get recovery support from macro and micro-nutrient depletion that occurs during pregnancy and lactation [8, 21, 25]. Therefore, an optimal birth interval is required for the protection of both mothers and babies [26–28].

Birth interval is a major indicator of fertility and it may have an influence in reducing the size of the country's population through fertility control [2, 29, 30]. As fertility rate of a country is inextricably related to the role of women, the status of women consisting of several indicators undoubtedly influences the population growth of a country. Several studies have shown that maternal education has a positive relationship with a country's population size [31, 32]. Moreover, the economic status of women is also related to fertility; the higher the economic status of women, the lower the fertility rate [33]. However, women empowerment could be another pathway to assess fertility preferences because empowered women can enhance their ability by expanding resources that can have control over achieving their ideal number of children [34, 35]. Women's education, empowerment, and economic status had already been proven to influence maternal health care utilization in developing countries [36, 37] and to be the major determinants to boost up vaccination uptakes in Pakistan [38]. A study conducted in Bangladesh found a significant association between number of living children and status of women namely level of education, occupation, and having discussion with their partners regarding family planning issues [39]. In this study, women's empowerment, education, and

economic status are jointly introduced as women's 3E which has been explored under the area of fertility to investigate the direction and magnitude of their association with birth interval.

A study using pooled data obtained from three Bangladesh Demographic and Health Surveys (BDHS) of the years 2004, 2007, and 2011 data, revealed that parity, survival status of index child, father's education, wealth quintile, place of residence, division, and year of the survey had significant effects on birth interval [40]. Research work conducted in the country analyzed determinants of birth interval using BDHS, 2007 data and showed that previous birth interval, mother's education, working status, and mass media exposure had a significant association with birth intervals [41]. Another study conducted by Mahmood and Zainab, utilizing the BDHS 2004 data, found that mother's education and place of residence had a great influence on variation in birth intervals in Bangladesh [42]. However, a study in the Philippines revealed a strong positive association between longer birth interval and women's decision-making autonomy [43].

Addressing a number of social needs, the Sustainable Development Goals (SDGs) aim to ensure a peaceful, sustainable, prosperous, and equitable life for all in the world by 2030. Women's 3E is directly linked to SDG1 for poverty alleviation, SDG4 for ensuring equitable quality education, and SDG5 for women's empowerment and gender equality [44]. Again, women's empowerment, education, and economic status can make a significant contribution to the achievement of many other SDGs, and therefore it is time to address these factors for maternal and child health outcomes in achieving the SDGs. According to the report of the World Bank, it was demonstrated that global development would be attained through gender equality [45]. Therefore, it is very crucial to study the importance of women's 3E in different aspects that can help policymakers to build up new strategies to ensure economic growth by promoting gender equity.

Although the total fertility rate (TFR) in Bangladesh has been 2.3 births per woman since 2011, the median birth interval has increased from 47.4 to 55.7 months over the last seven years [46, 47]. Despite this increase, TFR is still far from the goal of the health sector program to reach the target of 2 births per woman by 2022 [46]. Again, approximately 32%, 29%, and 25% of births were born at below three years of birth spacing in 2011, 2014, and 2017–18, respectively, where the results indicate a slow decline in the overall prevalence of short birth intervals [46–48]. Therefore, Bangladesh is still facing the problem of birth spacing to achieve the WHO recommendation that every mother should give subsequent birth at or after 33 months. Hence, it is necessary to accelerate the duration of birth interval in Bangladesh through necessary policy-making measures. Since birth interval can affect fertility, maternal and child health, it is a matter of great concern for researchers and policymakers to study the factors that affect birth interval in overpopulated countries like Bangladesh. The current study hypothesizes that 3E of women may be a necessary target for policy-making interventions to reduce subsequent births at short intervals so that the country can achieve the desired fertility rate.

In this study, an attempt has been made to investigate the association of women's 3E with short birth interval in Bangladesh controlling other important variables related to women, child, and household characteristics. To the researchers' knowledge, no study has been conducted in Bangladesh on this topic so far. For the purpose of analysis, secondary data extracted from BDHS, 2017–18 has been analyzed using the sequential approach of the logistic regression model.

## Materials and methods

### Study design

The study utilized the birth interval data extracted from the nationally representative Bangladesh Demographic and Health Survey (BDHS), 2017–18 data. The survey had been conducted

following a two-stage stratified sample of households. In the first stage, the survey selected 675 enumeration areas (EAs), 227 EAs in urban areas, and 448 in rural areas, with probability proportional to EA size and with an independent selection within each stratum. Afterward, 30 households from each of the selected EAs have been chosen in the second stage resulting in a total of 20,250 residential households, but a total of 20,160 households were selected for the survey in 672 clusters after elimination of three clusters that were completely eroded by floodwater. Finally, BDHS successfully interviewed 20,127 ever-married women aged 15–49 years on a complete history of their live births along with some important socio-economic and demographic characteristics. The details of the sampling design can be found in the BDHS, 2017–18 final report [46].

## Participants

To get information for the current study, at least one birth for current pregnant mothers and at least two births for non-pregnant mothers born in the 5 years before the survey have been considered which results in a total of 5,441 births. The selected births have come from a total of 4,866 reproductive women in the study. The details of sample selection are given in Fig 1. It is noted that the current pregnancies were treated as births and the duration between date of last birth and initiation of pregnancy is considered as birth interval.

## Ethics approval and consent to participate

The collection of demographic and health survey data for 2017–18 BDHS was approved by the Institutional Review Board of ICF International, Rockville, MD, USA and Bangladesh Medical Research Council, Dhaka, Bangladesh. The 2017–18 BDHS was implemented under the authority of the National Institute of Population Research and Training (NIPORT) of the Government of the People's Republic of Bangladesh with financial support from USAID/Bangladesh. Informed consent was obtained from all respondents to the survey prior to questioning. The survey excluded those respondents who did not consent.

## Outcome measure

The current study aimed to measure the association of women's 3E with short birth interval in Bangladesh. Therefore, the short birth interval was considered as a binary outcome variable in the study, where it takes the value 1 when the birth interval is below 33 months between two consecutive live births or fewer than 24 months for last birth to conception of a pregnancy following WHO guidelines on birth spacing [1].

To compute the birth intervals from the data extracted from BDHS, 2017–18, following two steps have been considered:

1. Currently non-pregnant mothers having at least two live births born in the preceding five years of survey were selected. Then birth intervals were measured taking the duration (in months) between two consecutive live births.

2. Currently pregnant mothers having at least one live birth born in preceding five years of survey were selected. For the purpose of computation of birth interval for the current pregnancy, duration (in months) between date of last birth and initiation of pregnancy has been considered and for all other births, birth intervals are computed by taking the duration (in months) between two successive births.

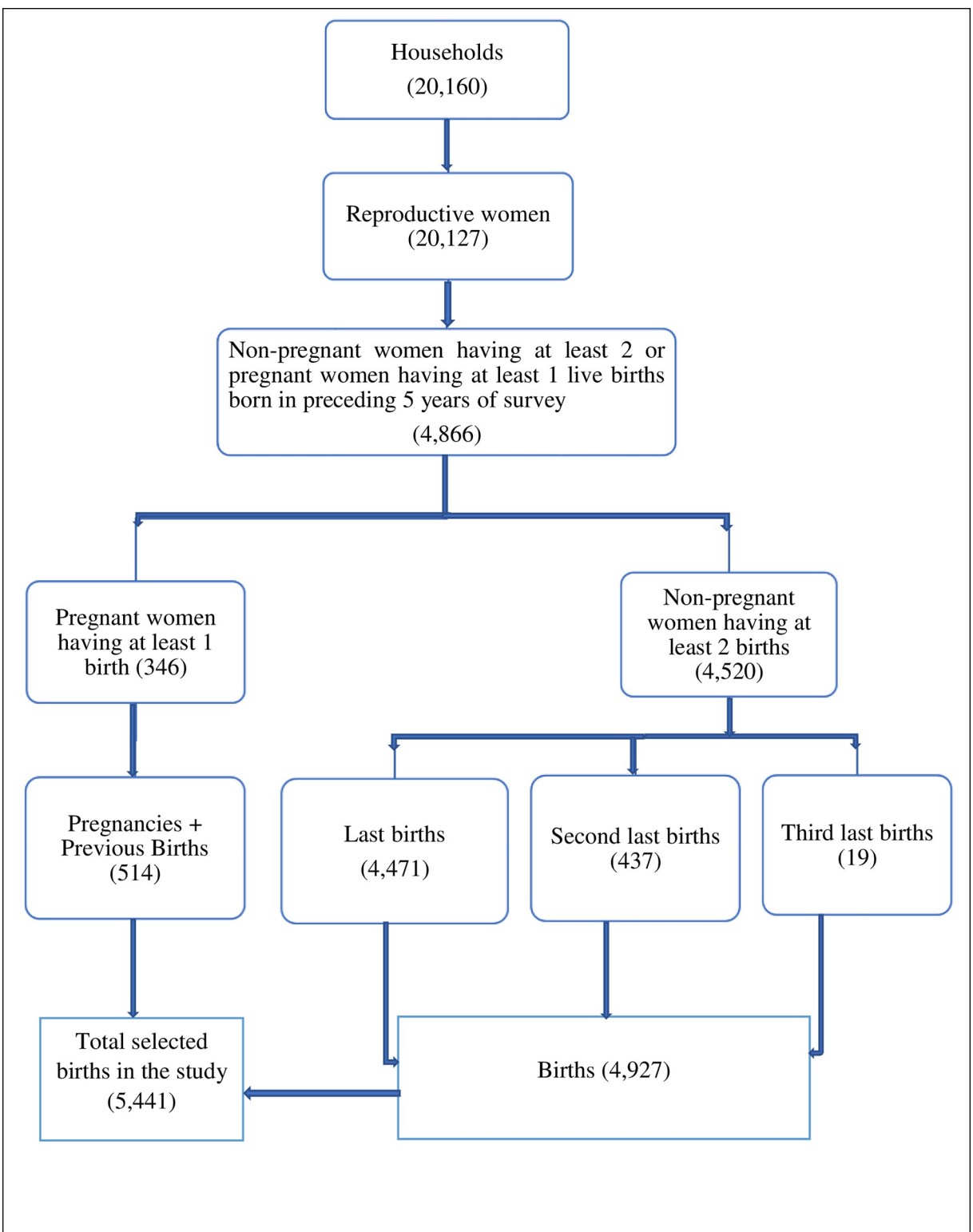

**Fig 1. Flow chart to select live births in computing birth intervals.**

## Main explanatory variable: Women's 3E index

This study emphasized understanding the short birth interval with respect to women's coverage in 3E. Women's empowerment, education, and economic status were jointly treated as women's 3E. Women's empowerment cannot be measured directly due to its latent phenomena. As a result, different sets of indicators have been used in different studies to define women's empowerment. Among those different indicators, women's participation in household decision making is estimated to be the most common that has been used in 37 studies during the year 1990 to 2012 [49]. In this study, women's empowerment has been assessed using the variables regarding their participation in household decision making, available in the DHS survey. These variables include respondent's involvement in decisions on (i) spending her own earning, (ii) spending her partner's earning, (iii) her own health care, (iv) large household purchases, and (v) visits to family relatives. Finally, these five variables are converted into dichotomous variables, where a value of 1 is given among those women who decided alone or in partnership with their husbands/partners, and a value of 0 for otherwise. Again, the principal component analysis (PCA) has been applied to create the wealth score index using the available household assets set by the survey [46], and then the score is divided into two equal parts based on the median to define economic status as poor and rich. Therefore, the economic status is an indicator variable giving a value of 1 for rich and 0 for poor women. The women's education is re-categorized into two categories to define indicator variable: uneducated (no and primary) coded as 0 and educated (secondary and higher) coded as 1. To obtain the joint influence of empowerment, education, and economic status, women's 3E index has been measured by computing the mean score of the indicator variables following WHO approach [50]. The approach follows a rule to give equal weights in each domain (empowerment, education, and economic status), and also to give equal weights in each indicator variable of the weighted domains to calculate the total mean score. For clear understanding, mathematically it can be written as

$$Z_i = \left( \frac{1}{k} \sum_{j=1}^{k} \frac{1}{h_j} \sum_{l=1}^{h_j} x_{ijl} \right) \times 100; i = 1, 2, \ldots, n; j = 1, 2, \ldots, k; l = 1, 2, \ldots, h_j,$$

where $Z_i$ is the total mean score of $i^{th}$ case, $k$ is the number of domains, $h_j$ is the number of indicators in $j^{th}$ domain, $x_{ijl}$ is the value of $l^{th}$ indicator in $j^{th}$ domain for $i^{th}$ case, and $n$ is the total number of cases [50, 51]. In the study, there are five indicators for empowerment and one for both education and economic status to calculate the total mean, which is the score of women's 3E for each woman. In the study, the scale reliability coefficient (Cronbach's $\alpha$) has also been calculated, and it was found to be of 0.73. Therefore, the result indicates that the selected seven indicators are internally related to each other and hence there is no problem in creating a score index [52]. To explore the effect of women's 3E on birth spacing at different levels of the score, we classified the score into four categories as below 50%, 50% - 75%, 75% - 100%, and full coverage in 3E (100%). Under full coverage of women's 3E, a woman is fully empowered, educated, and economically prosperous.

## Covariates

Based on a number of literatures [2, 6, 40–43, 53–63], the control variables chosen in this study are women related variables: age at marriage, access to media, employment status, husband's education and experience of terminated pregnancy; preceding child related variables: gender, birth order and survival status; household related variables: migration status, region and place of residence. The definition and measurements of the variables are given in Table 1.

## Statistical analysis

To get the univariate results for the selected variables, the frequency percentage has been used in the study. In bivariate analysis, the chi-square test with cross-tabulation was used to evaluate the unadjusted association between the selected variables and the short birth interval. To find the unadjusted as well as adjusted association of woman's 3E with short birth interval, after controlling the relevant covariates, a sequential approach of logistic regression model is used: (a) Model I: classified mean score of women's 3E; (b) Model II: Model I + women related variables; (c) Model III: Model II + preceding child related variables; and (d) Model IV: Model III + household related variables. Covariates having p-value less than 0.2 in the bivariate analysis were considered in the regression models [2]. The likelihood ratio test (LRT) has also been used to check the goodness of fit for the fitted models [64]. To analyze the data, STATA version 14 has been used in the study.

## Results

### Descriptive statistics

Table 2 illustrates the frequency percentages and mean scores of the indicators of women's 3E. Among the indicators of women's empowerment, a high proportion in visiting family relatives, taking their own health care, and a low proportion in spending their own earnings were found. More than one-third (35.31%) of selected women can decide on their own earnings, and about three-fourth can take the decisions on their own health care (75.83%) or family visits (74.31%) alone or jointly with their husbands. The results imply that most Bangladeshi women are empowered to make their own healthcare decisions, but women are less empowered to make decision in spending their own earnings. Again, at present, more than 50 percent of Bangladeshi women are educated and about 50% of women fail in the rich economic category. To compute the score of women's 3E, three domains are considered, which are women's empowerment consisting of five indicators, education and economic status. The mean scores for these domains were found to be 21.8, 18.8, and 16.7, respectively. Hence the total mean score of women's 3E was 57.3. It indicates that on average, a woman still requires 42.7 points to reach the full coverage of women's 3E.

Basic background characteristics of the respondents along with the percentage frequency distributions have been presented in Table 3. Table 3 also displays the percentage of short birth interval in each category of selected variables with 95% confidence interval as well as p-values obtained from chi-square test which signifies the unadjusted association of the variables with short birth interval.

The key independent variable in this study is women's 3E. From Table 3, it is observed that more than one-third of the respondents (39.1%) have coverage of below 50% in achieving women's 3E. However, 30.9% and 23.9% of women are found to have coverage 50%-75% and 75%-100%, respectively. It has been also found that only 6.0% of women have full coverage in this regard. Table 3 also reveals that more than half (56.5%) of the women have illiterate husbands. Moreover, more than two-thirds (72.9%) of the respondents got married off at or before the age of 18 years, while only 27.1% of them were over 18 years at the time of their marriage. It has been also found that 20.8% of the women had experienced terminated pregnancy in their lifetime, whereas, majority of them had not confronted such experiences. Furthermore, 40.2% of mothers are not exposed to mass media like radio, television, or newspaper, whereas 59.8% have access to them. Most of the respondents (92.6%) in the sample belong to the Muslim religion, while a few of them (7.4%) are non-Muslims. Results from Table 3 also show that 19.2% of the women have a positive attitude towards intimate partner violence, whereas 80.8%

**Table 1. Definition and measurements of the variables used in the study based on the 2017–18 BDHS.**

| Characteristics | Variables | Measurement |
|---|---|---|
| Birth Spacing | Birth Interval | Duration of months between two successive live births or birth to successive pregnancy |
| | Short Birth Interval | Birth Interval of below 24 months for birth-to-pregnancy or fewer than 33 months for birth-to-birth interval (Categories: Yes/No) |
| **Women's 3E Index | Women's Empowerment | Women's empowerment has been assessed using the variables regarding their participation in household decisions. |
| | Women's Education | The survey created four categories of women's education: no, primary, secondary, and higher. In the study, the education is converted into two categories: uneducated (no and primary) and educated (secondary and higher) |
| | Economic Status | The principal component analysis (PCA) has been applied to create the wealth score index using the available household assets set by the survey [46], and then the score is divided into two equal parts based on median to define economic status as poor and rich. |
| Women | Husband's Education | Following the conversion of women's education, the current study also created the husband's education in two categories: uneducated and educated |
| | Maternal Age at Marriage | Age at marriage is categorized into three categories as below 15 years, 15–17, and 18 or 18+ years. |
| | Pregnancy Termination | Pregnancy termination is defined as ending of a pregnancy which results in no live birth. The study used this variable as whether the women ever terminated a pregnancy or not before the survey. |
| | Employment Status | Yes/No |
| | Media Exposure | Media exposure is created based on variables of watching TV, listening radio, and reading newspapers/magazine, and a woman is exposed to media who has exposure to either of the media sources (Categories: Yes/No) |
| | Religion | The original dataset has four categories of religion: Islam, Hinduism, Buddhism, and Christianity, but the religion has been re-categorized into two categories as Muslim (Islam) and Non-Muslim (Hinduism, Buddhism, and Christianity) in the study. |
| | Intimate Partner Violence | Intimate partner violence has been measured using the five proxy variables of wife beating: (i) beating justified if wife goes out without telling husband (ii) beating justified if wife neglects the children (iii) beating justified if wife argues with husband (iv) beating justified if wife refuses to have sex with husband and (v) beating justified if wife burns the food. A woman is treated as violated by her partner/husband if she faces in any of the above beatings. Categories: Yes/NO |
| Child | Gender of Preceding Child | Gender is considered for preceding Child in the study. Categories: Male/Female |
| | Birth Order of Preceding Child | Categorized into three categories as first birth, second birth, and third/higher birth |
| | Survival of Preceding Child | Yes/No |
| Household | Place of Residence | Urban/rural |
| | Region | The original dataset has eight divisions, but the study divided whole Bangladesh into three regions using divisions: Central (Barisal, Mymensingh and Dhaka divisions), Eastern (Chattogram and Sylhet divisions), and Western (Khulna, Rajshahi and Rangpur divisions). |
| | Migration Status | Yes/No |

Note.

**Women's 3E index has been measured using the variables of women's empowerment, education and economic status, where the score of women's 3E is computed following WHO approach [50]

of them have not. Moreover, it is also found that 45.2% of women are employed, while 54.8% are not. However, the proportions of male and female index children are almost equal in the sample. For the case of birth order number of preceding child, it can be observed that most of the children (53.1%) in the sample are the first births of their mothers, whereas, 27.1% and 19.9% of them have birth order number two, and greater than two, respectively. It has also been specified that 6.3% of children who are older between two consecutive births have experienced death, while 93.7% are alive. Again, two-thirds of the respondents reside in rural areas, whereas, the rest of them belong to urban areas. However, the sample includes 89.0% of the women who are migrants, while 11.0% are not. In addition, 33.5%, 36.4%, and 30.0% of the respondents are from the eastern, central, and western regions of Bangladesh, respectively.

**Table 2. Frequency distribution of several indicators of women's 3E along with its mean score and 95% confidence intervals.**

| | Indicators in computing women's 3E | | | | | | | Mean score of women's 3E |
|---|---|---|---|---|---|---|---|---|
| | *Women's empowerment | | | | | Education Status | Economic Status | |
| | *Decided on spending her own earnings* | *Decided on spending her partner's earnings* | *Decided on her own health care* | *Decided on large household purchases* | *Decided on visits to family relatives* | *Educated* | *Rich* | |
| Percentage | 35.31 | 69.16 | 75.83 | 72.74 | 74.31 | 56.39 | 49.97 | |
| Mean score (in %) | 2.4 [2.3–2.4] | 4.6 [4.5–4.7] | 5.1 [5.0–5.1] | 4.8 [4.8–4.9] | 5.0 [4.9–5.0] | 18.8 [18.4–19.2] | 16.7 [16.2–17.1] | |
| Total | 21.8 [21.5–22.1] | | | | | 18.8 [18.4–19.2] | 16.7 [16.2–17.1] | 57.3 [56.5–58.0] |

Note

* Women participated in household decisions alone or in partnership with their husbands/partners

Table 3 reveals that 22.5% of birth intervals are in the short interval. The prevalence of short birth interval is highest (27.1%) among the women who possess less than 50% coverage of women's 3E. An expected direction between short birth interval and coverage levels of women's 3E has been observed that women with a higher coverage level of 3E tend to have a lower chance of having a short birth interval. Results show that the percentage of short birth interval for the women with coverage of 50%-75%, 75%-100%, and 100% are 21.4, 18.3, and 14.9, respectively. However, the p-value<0.001 presented in Table 3 provides enough evidence to justify women's 3E to be a powerful tool that plays a significant role to control the risk of having a short birth interval.

Table 3 also discloses the percentages of short birth intervals along with their 95% confidence intervals for the categories of other selected variables. Results from p-values confirm that the variables husband's education, maternal age at marriage, pregnancy termination, media exposure, employment status, birth-order of preceding child, survival of preceding child, place of residence, and region have a significant unadjusted association with short birth interval as the p-values are less than 0.10.

## Sequential logistic regression models

The study analyzes the data using the sequential approach of the logistic regression model in order to find out the association of women's 3E with short birth interval in Bangladesh. To serve this purpose, four separate models have been entailed where the first model (Model I) involves only women's 3Es as a covariate. The second model (Model II) considers some women related variables along with women's 3E to examine how these women related variables change the magnitude of association between women's 3E and short birth interval. Similarly, the third (Model III) model extends Model II by incorporating some variables related to child characteristics. Lastly, the final model (Model IV) encompasses all covariates from Model III with the addition of some household related covariates. For all the models, first, we check the overall goodness of fit by likelihood ratio test (LRT) based on chi-square distribution and then we examine the significance of the included covariates. The LRT provided that all the models were fitted well as the p-values were below 0.001. Table 4 presents the estimates of the parameters, odds ratios along with confidence intervals and p-values for all the four models.

Results from Table 4 for Model I confirm a significant unadjusted association of women's 3E with short birth spacing. It has been observed that the odds of having a short birth interval decreases by 26%, and 39% for women who have 50% -75%, and 75%- 100% coverage of

**Table 3. Frequency percentage, and short birth interval with 95% confidence intervals (CI) by the selected background characteristics.**

| Characteristics | Frequency Percentage | Short Birth Interval (95% CI) | p-value |
|---|---|---|---|
| **Women's 3E** | | | |
| Coverage of < 50% | 39.1 | 27.1 [25.2–28.9] | |
| 50% - 75% | 30.9 | 21.4 [19.5–23.4] | <0.001 |
| 75% - 100% | 23.9 | 18.3 [16.2–20.1] | |
| Full Coverage (100%) | 6.0 | 14.9 [11.1–18.8] | |
| **Husband's Education** | | | |
| Uneducated | 56.5 | 24.3 [22.7–25.8] | <0.001 |
| Educated | 43.5 | 20.2 [18.6–21.8] | |
| **Maternal Age at Marriage** (in years) | | | |
| 10–15 | 28.0 | 19.1 [17.1–21.0] | |
| 15–18 | 44.9 | 22.5 [20.8–24.1] | <0.001 |
| 18+ | 27.1 | 26.1 [23.9–28.4] | |
| **Pregnancy Termination** | | | |
| No | 79.2 | 23.1 [22.8–24.3] | 0.053 |
| Yes | 20.8 | 20.4 [18.0–22.7] | |
| **Media Exposure** | | | |
| No | 40.2 | 25.0 [23.2–26.8] | <0.001 |
| Yes | 59.8 | 20.8 [19.4–22.2] | |
| **Religion** | | | |
| Muslim | 92.6 | 23.8 [21.6–23.9] | 0.111 |
| Non-Muslim | 7.4 | 19.3 [15.4–23.2] | |
| **Intimate Partner Violence** | | | |
| No | 80.8 | 22.4 [21.2–23.6] | 0.761 |
| Yes | 19.2 | 22.8 [20.3–25.4] | |
| **Employment Status** | | | |
| No | 54.8 | 24.8 [23.3–26.4] | <0.001 |
| Yes | 45.2 | 19.7 [18.1–21.2] | |
| **Gender of Preceding Child** | | | |
| Male | 48.5 | 22.7 [21.1–24.3] | 0.778 |
| Female | 51.6 | 22.3 [20.8–23.9] | |
| **Birth Order of Preceding Child** | | | |
| First Birth | 53.1 | 21.9 [20.4–23.4] | |
| Second Birth | 27.1 | 20.3 [18.3–22.4] | <0.001 |
| Third/Higher Birth | 19.9 | 26.9 [24.3–29.6] | |
| **Survival of Preceding Child** | | | |
| No | 6.3 | 56.9 [51.6–62.1] | <0.001 |
| Yes | 93.7 | 20.2 [19.1–21.3] | |
| **Place of Residence** | | | |
| Rural | 66.7 | 23.3 [21.9–24.6] | 0.057 |
| Urban | 33.3 | 21.0 [19.1–22.8] | |
| **Migration Status** | | | |
| No | 11.0 | 21.7 [18.4–25.1] | 0.639 |
| Yes | 89.0 | 22.6 [21.4–23.8] | |
| **Region** | | | |
| Eastern | 33.5 | 30.9 [28.7–33.0] | |
| Central | 36.4 | 19.9 [18.1–21.6] | <0.001 |
| Western | 30.0 | 16.3 [14.5–18.1] | |
| **Total** | **n = 5,441** | **22.5 [21.4–23.6]** | |

**Table 4. Odds ratio (OR) with 95% confidence intervals (CI) obtained from sequential binary logistic regression models.**

| Characteristics | OR with 95% CI | | | |
|---|---|---|---|---|
| | Model I | Model II | Model III | Model IV |
| **Women's 3E** | | | | |
| Coverage of < 50% | 1.00 | 1.00 | 1.00 | 1.00 |
| 50% - 75% | 0.74*** [0.63–0.86] | 0.74*** [0.63–0.87] | 0.76** [0.65–0.90] | 0.77** [0.65–0.91] |
| 75% - 100% | 0.61*** [0.51–0.72] | 0.55*** [0.45–0.67] | 0.56*** [0.46–0.69] | 0.59*** [0.48–0.73] |
| Full Coverage (100%) | 0.47*** [0.34–0.65] | 0.55** [0.39–0.77] | 0.56** [0.39–0.80] | 0.58** [0.41–0.83] |
| **Maternal Age at Marriage** (in years) | | | | |
| 10–15 | | 1.00 | 1.00 | 1.00 |
| 15–18 | | 1.31** [1.12–1.54] | 1.38*** [1.17–1.63] | 1.25** [1.05–1.48] |
| 18+ | | 1.73*** [1.45–2.08] | 1.84*** [1.52–2.21] | 1.54*** [1.27–1.86] |
| **Husband's Education** | | | | |
| Uneducated | | 1.00 | 1.00 | 1.00 |
| Educated | | 0.90 [0.78–1.05] | 0.93 [0.79–1.08] | 0.93 [0.80–1.08] |
| **Pregnancy Termination** | | | | |
| No | | 1.00 | 1.00 | 1.00 |
| Yes | | 0.86+ [0.74–1.02] | 0.84* [0.71–0.99] | 0.85+ [0.72–1.01] |
| **Media Exposure** | | | | |
| No | | 1.00 | 1.00 | 1.00 |
| Yes | | 0.93 [0.80–1.07] | 0.93 [0.81–1.08] | 0.98 [0.85–1.14] |
| **Religion** | | | | |
| Muslim | | 1.31* [1.01–1.70] | 1.35* [1.03–1.77] | 1.40* [1.06–1.83] |
| Non-Muslim | | 1.00 | 1.00 | 1.00 |
| **Employment Status** | | | | |
| No | | 1.00 | 1.00 | 1.00 |
| Yes | | 0.72*** [0.63–0.83] | 0.71*** [0.61–0.81] | 0.80** [0.69–0.92] |
| **Birth Order of Preceding Child** | | | | |
| First Birth | | | 1.00 | 1.00 |
| Second Birth | | | 0.89 [0.76–1.05] | 0.85 [0.72–0.99] |
| Third/Higher Birth | | | 1.16* [0.97–1.38] | 1.03 [0.86–1.23] |
| **Survival of Preceding Child** | | | | |
| No | | | 1.00 | 1.00 |
| Yes | | | 0.19*** [0.15–0.24] | 0.18*** [0.15–0.23] |
| **Place of Residence** | | | | |
| Rural | | | | 1.00 |
| Urban | | | | 0.96 [0.83–1.12] |
| **Region** | | | | |
| Eastern | | | | 1.69*** [1.44–1.98] |
| Central | | | | 1.00 |
| Western | | | | 0.86 [0.73–1.04] |
| Log Likelihood | -2875.38 | -2840.36 | -2734.11 | -2701.12 |

Note

+p < 0.10

*p < 0.05

**p < 0.01

***p < 0.001.

women's 3E, respectively compared to those with coverage level less than 50%. Again, women with full coverage of 3E have 53% lower odds of experiencing a short birth interval compared to women having <50% coverage of 3E.

Results obtained from Model II show a significant negative relationship between women's 3E and short birth interval. It has been found that mothers who assure 50% -75%, 75%- 100%, and 100% coverage of women's 3E respectively, are 26%, 45%, and 45% less likely to experience a short birth interval compared to those with 50% or lower coverage of 3E. Result also confirms maternal age at marriage, pregnancy termination, religion, and employment status to have a significant association with short birth interval. It has been found that mothers whose age at marriage is 15–18 and 18+ have 31% and 73% higher odds of having a short birth interval, respectively compared to the women aged between 10 to 15 years at the time of their marriage. Again, mothers who experienced termination of pregnancy have 14% lower odds of having a short birth spacing compared to those who did not. Moreover, Muslim women are 31% more likely to deal with short birth spacing compared to non-Muslim women. Furthermore, employed mothers have 28% lower odds of giving birth with a short interval compared to unemployed mothers. However, Model II also exhibits that the husband's education level and media exposure have no significant association with birth interval.

The association between women's 3E and length of birth interval has been analyzed in Model III after adjusting for some women, and child related variables. Likewise, the previous two models, Model III also provides enough evidence to justify a significantly strong association of women's 3E and short birth interval. It has been observed that the odds of experiencing a short birth interval reduces by 24%, 44%, and 44%, when mothers possess 50% -75%, 75%-100%, and 100% coverage of women's 3E, respectively. According to Model III, there are 38%, and 84% higher odds of having a short birth interval for women who got married at the age of 15–18, and 18+ years compared to their counterparts with age at marriage of 10–15 years. Moreover, the odds of short birth interval for mothers who experienced terminated pregnancies reduce by 16% compared to mothers who had no such experiences. Again, Muslim women have 35% higher odds of short birth spacing compared to non-Muslim women. Furthermore, employed mothers are found to have 29% lower odds of a short birth spacing compared to unemployed mothers. In addition, children with birth order number four or more are likely to have 16% higher odds to be born with a short birth interval compared to the subsequent baby of the first child. However, for a mother having a previous child alive, the odds of short birth interval is 81% lower compared to a mother having lost the previous child. Likewise, Model II, Model III does not show any significant association of husband's education and media exposure with short birth interval.

Model IV synthesizes all the covariates considered in model III with an addition of women's household characteristics. After controlling those important women, child, and household related variables, women's 3E remains consistent to retain a significant association with short birth interval. Results reveal that the odds of short birth interval decreases by 23%, 41%, and 42% with women's 50%-75%, 75%-100%, and 100% coverage level of 3E, respectively compared to women with less than 50% coverage. That is women's 3E plays a vital role to minimize the risk of short birth interval. Moreover, women who got married at the age of 15–18, and 18 + years have respectively 25%, and 54% higher odds of short birth interval than those aged 10–15 years. Moreover, mothers who experienced a terminated pregnancy in their lifetime have 15% lower odds of short birth interval than mothers who did not. Furthermore, women who belong to Muslim families are supposed to have 40% higher odds of occurring short birth interval than non-Muslim women. Again, employed mothers are 20% less likely to experience a short birth interval compared to unemployed mothers. Moreover, mothers having the previous child alive have 82% lower odds of short birth interval compared to those with a dead

child. In addition, women who reside in eastern and western regional areas of Bangladesh have respectively, 69% higher, and 14% lower odds of short birth interval compared to mothers who originated from the central areas of Bangladesh. However, husband's education, media exposure, birth order number of preceding child, and place of residence have no significant association with birth spacing.

## Discussion

This study showed that the overall prevalence of short birth interval is 22.5 in recent years. That is, more than one in every 5 children is born with a short birth interval in Bangladesh. This finding is consistent with the findings from Rwanda (20%), Cameroon (21.3%), and Uganda (25.3%) [65]. Compared to Bangladesh, the prevalence of short birth interval is higher in Iran (28.5%) and Ethiopia (45.8%) [54, 60]. Previous study of Bangladesh using data from BDHS, 2014 found the prevalence of 24.6%, which suggests a poor declining trend in short birth spacing compared to the current study [6]. However, this is a matter to be concerned about because those children born with a short birth interval are likely to experience early neonatal death [66]. Moreover, mothers who gave births before the recommended optimal birth interval carry an increased risk of developing pregnancy and health complications [66]. Although the proportion of short birth interval has decreased since 2014, our desired TFR is yet to be achieved.

Over the last two decades, researchers have paid much attention to study women's status focusing extra care towards women's empowerment, educational level, and economic status because of their significant contribution to different aspects of development including reproductive health, worldwide. However, the study aimed to probe into the influence of the power of these three socio-economic factors together, presented in this paper as women's 3E index, on the current scenario of short birth interval in Bangladesh. To serve this purpose, a secondary data, extracted from BDHS, 2017–18 data, have been analyzed where the births of reproductive women preceding five years of the survey are considered excluding those births of non-pregnant mothers who have one child. In the bivariate analysis, the unadjusted associations of women's 3E have been analyzed and the result signifies a strong association between short birth interval and 3E. It has been observed that the proportion of having short birth interval is higher for women who have less coverage in achieving 3E. In regression analysis, sequential approach of logistic regression model has been utilized to find out the adjusted association of women's 3E index with short birth spacing and to observe how these relationships behave while other important covariates related to women, child, and household characteristics acted sequentially to the model. Findings show that at each step, women's 3E remains a significant factor with similar direction and magnitude and thereby, contributes significantly to control over the choice of birth interval.

Analyzing the four models, it has been concluded that women's 3E index has a strong impact on the practice of short birth interval in Bangladesh. Hence, women empowerment, education, and wealth index can be jointly considered as a potential force to influence the likelihood of having a short birth interval. However, women empowerment has the highest contribution to ensure the 3E of women, as reported in this study. Moreover, female education and economic status have almost the same contribution to women's 3E index. Therefore, these three indicators can be considered as influential determinants of short birth interval. Results reveal a negative association between women's 3E and short birth interval. That is, a woman is less likely to experience a short birth interval when she is exposed to the higher coverage of women's 3E. Hence, women who are empowered, educated, and have higher economic status are less likely to experience a short birth interval. To be specific, women empowerment has a

negative association with short birth interval. This finding is in tune with some previous works conducted in other countries [43]. Therefore, this is the high time to ensure women to be more empowered because according to the study, empowered women have access to various household-related decisions and thereby, can involve themselves in decisions regarding birth interval behaviors to some great extent. Some studies proposed that in less developed countries, lower fertility rates can be achieved by enhancing gender equity [67, 68]. This is because the magnitude to which women start practicing equal rights with men, can contribute their opinion with freedom regarding opting different family planning programs which in return helps in deciding desirable family size.

Educating women is another pivotal approach to alleviate the risk of short birth interval by means of attaining the power of women's 3E, as demonstrated in this study. Some other studies also figured out the importance of educating women to minimize the length of birth interval [40–42, 69]. This could be because of the well-established fact that educated women are more likely to have the knowledge regarding adverse health outcomes of short birth spacing and therefore they are conscious enough of using modern contraception to lengthen their birth intervals than uneducated women [57]. At the same time, the higher education level of women is associated with getting better job opportunities which leads women to work outside their homes [70]. As a result, it may let women choose a limited number of children which consequently lengthens spacing between births.

According to the study, women with higher economic status act positively to attain greater coverage by assuring 3E of women. Therefore, it can be concluded that the likelihood of having a short birth interval is lower for women who are economically rich which is consistent with the findings of some previous studies [60, 69, 71]. A recent study conducted under some rural areas in Bangladesh also showed that socio-economically disadvantageous women are more likely to have a short birth interval [6]. The reason behind this finding could probably be the fact that the access to health knowledge, family planning, and maternal health care services for women is getting enhanced with the increase in their economic status. Besides it was demonstrated by a recent study that the higher socioeconomic status is associated with the increased use of modern contraceptives which consequently reduces the chance of having the next child and thereby lengthens the spacing between two successive births [72]. In addition, the study has brought some secondary findings. The maternal age at marriage, experience of terminated pregnancy, religion, employment status, survival status of previous child, and region are found to have a significant adjusted association with birth interval.

## Limitations

This study utilized the cross-sectional data extracted from BDHS, 2017–18 data which restricts the scope to analyze the causal relationship between women's 3E and short birth interval. Moreover, the study did not take into account some other important control variables due to absence in data. In addition, recent births born in the five years prior to BDHS, 2017–18 have been considered that may create a recall bias in the study.

## Conclusion and recommendation

Women empowerment, education, and economic status can be jointly considered as a power of women's 3E, which is a vigorous element to play a highly significant role to control over the choice of birth interval, as evident from the study. Therefore, increasing the status of women by elevating educational attainment, economic conditions, and allowing them to actively participate in household decision-making would help lessen the fertility rate in Bangladesh. To sum up, in order to address the desired TFR and prevent intricacies of health outcomes for

both mother and child, an urgent need arises to initiate necessary strategies in this regard. These strategies should include programs for the expansion of women's educational opportunities and promotion of frequent family planning counseling to those who are economically less stable, and also to encourage women to feel free to discuss any fertility related issues, especially with their spouses. Although timing and spacing of pregnancy is a matter of choice for couples based on their personal preferences and situation, it is a responsibility for those involved in family planning, and maternal and child health care to convey the message of the optimal birth interval to general people to prevent adverse health and survival outcomes of the mother and newborn.

## Acknowledgments

We would like to thank the DHS Program and NIPORT, Bangladesh for allowing us to use the BDHS, 2017–18 data for the current study.

## Author Contributions

**Conceptualization:** Fatima Tuz-Zahura, Kanchan Kumar Sen, Shahnaz Nilima, Wasimul Bari.

**Data curation:** Fatima Tuz-Zahura.

**Formal analysis:** Kanchan Kumar Sen, Wasimul Bari.

**Investigation:** Fatima Tuz-Zahura, Shahnaz Nilima.

**Methodology:** Fatima Tuz-Zahura, Kanchan Kumar Sen, Wasimul Bari.

**Resources:** Shahnaz Nilima.

**Software:** Kanchan Kumar Sen.

**Supervision:** Wasimul Bari.

**Writing – original draft:** Fatima Tuz-Zahura, Kanchan Kumar Sen, Shahnaz Nilima.

**Writing – review & editing:** Fatima Tuz-Zahura, Kanchan Kumar Sen, Shahnaz Nilima, Wasimul Bari.

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
