## [Decision Letter · Decision Letter 0]

29 Jun 2021

PONE-D-21-15599

Can Women’s 3E Index Impede Short Birth Interval? Evidence from Bangladesh Demographic and Health Survey, 2017-18

PLOS ONE

Dear Dr. Zahura,

Thank you for submitting your manuscript to PLOS ONE. After careful consideration, we feel that it has merit but does not fully meet PLOS ONE’s publication criteria as it currently stands. Therefore, we invite you to submit a revised version of the manuscript that addresses the points raised during the review process.

The paper has been reviewed by three independent reviewers. The reviewers identified some important issues those need to be fixed before taking final decision. 

We look forward to receiving your revised manuscript.

Kind regards,

Enamul Kabir

Academic Editor

PLOS ONE

Journal Requirements:

Reviewers' comments:

Reviewer's Responses to Questions

**Comments to the Author**

1. Is the manuscript technically sound, and do the data support the conclusions?

Reviewer #1: Yes

Reviewer #2: Yes

Reviewer #3: Yes

2. Has the statistical analysis been performed appropriately and rigorously? 

Reviewer #1: Yes

Reviewer #2: No

Reviewer #3: Yes

3. Have the authors made all data underlying the findings in their manuscript fully available?

Reviewer #1: Yes

Reviewer #2: Yes

Reviewer #3: Yes

4. Is the manuscript presented in an intelligible fashion and written in standard English?

Reviewer #1: Yes

Reviewer #2: No

Reviewer #3: No

5. Review Comments to the Author

Reviewer #1: At a glance, the manuscript is technically sound and written in a knowledgeable manner. Introduction, results, discussion are presented wisely. Though required some minor modifications and corrections.

At first, with regard to the participants of the study, (line number 138, 139) and in outcome measure (line 156, 157) please co-relate the lines. In the outcome measure it is mentioned that, duration of current pregnancy has been taken into account instead of initiation of pregnancy or date of conception which was mentioned earlier in participants. As two of this are two different things, please check and correct the points. There are dissimilarities within the lines.

Secondly, Please specify the outcome measures more accurately, particularly which secondary outcome measures are specified for this study besides the primary outcome, not the variables.

Thirdly, which variables are used to determine wealth score index please mention specifically.

Fourthly, In conclusion, Currently, Bangladesh government has working extensively with family planning even at grass root level along with distribution of family planning instruments to people. Thereby, its solely a work of family planning health professionals, not all health care providers are responsible for this. Please rephrase your conclusion in this viewpoint with more preference on other strategies.

Reviewer #2: Comment #1: Abstract: Background: in the Abstract part well written and structured but in your paper abstract background part express a number of studies found the significant association of women’s 3E in maternal health outcomes, but no studies, to the best of knowledge, have been found to justify the joint influence of women’s 3E on the birth interval. As several studies have revealed that the short birth interval increases the risk of adverse maternal, perinatal and infant outcomes and it is also responsible for increasing the country’s population size, more research is needed on the birth interval. The above two paragraph is controversial and in your manuscript page four starting from paragraph four and the first paragraph of page five talks is several study including Bangladesh justify the joint influence of women’s 3E on the birth interval so, why you say there is no studies, to the best of knowledge, have been found to justify the joint influence of women’s 3E on the birth interval?

Comment #2: Abstract: Conclusion: In Conclusion part you recommended Policy-making interventions are needed to raise awareness among uneducated, under- empowered and economically poor reproductive women through family planning and fertility control programs so that the country can achieve the desired fertility rate, so say Conclusion and recommendation.

Comment #2: Method and Materials: Bangladesh Demographic and Health Survey (BDHS), 2017-18 data in Study design can you add reference of BDHS?

Comment #3: Method and Materials: table 1. Definition and measurements of the variables used in the study based on the 2017-18 BDHS: In the study, the education is converted into two categories: uneducated (no and primary), can we say primary education conclude as uneducated?

Comment #4: Method and Materials: what are your study participant’s inclusion and exclusion criteria?

Comment #5: Method and Materials: Statistical Analysis: what software uses to enter data and what software used to analysis data? Can you check goodness of fit tests? If you yes by what? You say Covariates having p-value less than 0.2 in the bivariate analysis were considered in the regression models. When you say statistically significant association independent and outcome variable? Please write detail in statistical analysis parts?

Comment #6: Method and Materials: Statistical Analysis: Explain how you went from OR to AOR, that is, what did you adjust for?

Comment #7: The English must be improved. There are numerous errors. Some words start with a capital letter for no reason. Spelling errors can be corrected using Word spell check.

Comment #9: Update references with the most recent

Reviewer #3: Thanks for give an opportunity to plos one chief editors to reviewing this article.

This research is crucial and appreciated to Empowerment, Education, Economic Status of Women in resource limited in general and Bangladesh in particular. Hence, the study has shown that the 3E in women can contribute in prolonging the duration of subsequent births in Bangladesh. Policy-making interventions are needed to raise awareness among uneducated, under-empowered and economically poor reproductive women through family planning and fertility control programs so that the country can achieve the desired fertility rate.

Comments:

Minor

1. General English: The review recommends the article be copyedited to improve language and grammar used

2. Better to add in introduction section about what has been done in Can Women’s 3E Index Impede Short Birth Interval? What are the identified gaps?

3. Conclusion so wide, just it should be based on your result only?

6. PLOS authors have the option to publish the peer review history of their article (what does this mean?). If published, this will include your full peer review and any attached files.

Reviewer #1: No

Reviewer #2: **Yes: **Dejen Getaneh Feleke

Reviewer #3: No

---

## [Author Response · Author response to Decision Letter 0]

16 Jul 2021

Journal requirements

According to journal requirements, data should be available without restrictions for further justification. The data used in our current study are fully available in the archives of DHS (Demographic and Health Survey) program. The DHS program handles all DHS data. We have downloaded the data after taking permission from DHS program. According to the instructions of the DHS program, data that we downloaded can only be analyzed for our requested research topic, and we are not allowed to share the downloaded data with others. However, anyone can download the dataset from DHS program after completing a simple registration process in the website www.dhsprogram.com.

Ethics approval and consent to participate

The collection of demographic and health survey data for 2017-18 BDHS was approved by the Institutional Review Board of ICF International, Rockville, MD, USA and Bangladesh Medical Research Council, Dhaka, Bangladesh. The 2017-18 BDHS was implemented under the authority of the National Institute of Population Research and Training (NIPORT) of the Government of the People's Republic of Bangladesh with financial support from USAID/Bangladesh. Informed consent was obtained from all respondents to the survey prior to questioning. The survey excluded those respondents who did not consent.

Response to reviewer 1

Comment 1: “At first, with regard to the participants of the study, (line number 138, 139) and in outcome measure (line 156, 157) please co-relate the lines. In the outcome measure it is mentioned that, duration of current pregnancy has been taken into account instead of initiation of pregnancy or date of conception which was mentioned earlier in participants. As two of this are two different things, please check and correct the points. There are dissimilarities within the lines.”

Response: Has been taken into account on Page 8 and 9.

Comment 2: “Please specify the outcome measures more accurately, particularly which secondary outcome measures are specified for this study besides the primary outcome, not the variables.”

Response: Have been specified on Page 8.

Comment 3: “which variables are used to determine wealth score index please mention specifically.”

Response: As we are using secondary data extracted from BDHS, 2017-18 data where the variable wealth index score had already been generated, we have added the reference in the manuscript to get detailed information about the variable on Page 10 and in Table 1.

Comment 4: “In conclusion, Currently, Bangladesh government has working extensively with family planning even at grass root level along with distribution of family planning instruments to people. Thereby, its solely a work of family planning health professionals, not all health care providers are responsible for this. Please rephrase your conclusion in this viewpoint with more preference on other strategies.”

Response: Has been taken care of on Page 29 in Line 470.

Response to reviewer 2

Comment 1: “Abstract: Background: in the Abstract part well written and structured but in your paper abstract background part express a number of studies found the significant association of women’s 3E in maternal health outcomes, but no studies, to the best of knowledge, have been found to justify the joint influence of women’s 3E on the birth interval. As several studies have revealed that the short birth interval increases the risk of adverse maternal, perinatal and infant outcomes and it is also responsible for increasing the country’s population size, more research is needed on the birth interval. The above two paragraph is controversial and in your manuscript page four starting from paragraph four and the first paragraph of page five talks is several study including Bangladesh justify the joint influence of women’s 3E on the birth interval so, why you say there is no studies, to the best of knowledge, have been found to justify the joint influence of women’s 3E on the birth interval?”

Response: Has been taken care of accordingly. Note that in literature, the variables women’s empowerment, education and economic status were used as covariates separately, whereas in this manuscript the joint effect of these three covariates was considered.

Comment 2: “Abstract: Conclusion: In Conclusion part you recommended Policy-making interventions are needed to raise awareness among uneducated, under- empowered and economically poor reproductive women through family planning and fertility control programs so that the country can achieve the desired fertility rate, so say Conclusion and recommendation.

”

Response: Has been adjusted accordingly.

Comment 2: “Method and Materials: Bangladesh Demographic and Health Survey (BDHS), 2017-18 data in Study design can you add reference of BDHS?”

Response: Reference was added.

Comment 3: “Method and Materials: table 1. Definition and measurements of the variables used in the study based on the 2017-18 BDHS: In the study, the education is converted into two categories: uneducated (no and primary), can we say primary education conclude as uneducated? 

Response: Topics related to maternal and child health care is not included in the curriculum of primary level of education in Bangladesh. Therefore, we considered primary education level in the uneducated group.

Comment 4: “Method and Materials: what are your study participant’s inclusion and exclusion criteria?”

Response: Inclusion and exclusion criteria have been provided in Figure 1.

Comment 5: “Method and Materials: Statistical Analysis: what software uses to enter data and what software used to analysis data? Can you check goodness of fit tests? If you yes by what? You say Covariates having p-value less than 0.2 in the bivariate analysis were considered in the regression models. When you say statistically significant association independent and outcome variable? Please write detail in statistical analysis parts?”

Response: The name of the Software has been added on Page 13. The goodness of fit test has also been performed (provided on Page 19) and result is included in Table 4. 

The reference has been included on Page 13 why p-value less than 0.2 was used. The p-value below 0.10 was used to define statistically significant association which was mentioned on Page 19.

Comment 6: “Method and Materials: Statistical Analysis: Explain how you went from OR to AOR, that is, what did you adjust for?”

Response: Note that in our manuscript, the term AOR was not used. The raised query in Comment 6 was addressed on Page 13 in Lines 216-217.

Comment 7: “The English must be improved. There are numerous errors. Some words start with a capital letter for no reason. Spelling errors can be corrected using Word spell check.”

Response: Has been taken care of. Microsoft word office 10 was used to prepare the draft.

Comment 9: “Update references with the most recent.”

Response: Has been updated.

Response to reviewer 3

Comment 1: “The review recommends the article be copyedited to improve language and grammar used.”

Response: Has been taken care of. We have used Microsoft Word 2010 version to write this manuscript.

Comment 2: “Better to add in introduction section about what has been done in Can Women’s 3E Index Impede Short Birth Interval? What are the identified gaps?”

Response: We have slightly modified the Introduction section to adjust this point.

Comment 3: “Conclusion so wide, just it should be based on your result only?”

Response: Since recommendations have been provided in conclusion section, the conclusion section was so wide. In revised version, we renamed this section as Conclusion and Recommendation.

---

## [Decision Letter · Decision Letter 1]

20 Sep 2021

PONE-D-21-15599R1Can Women’s 3E Index Impede Short Birth Interval? Evidence from Bangladesh Demographic and Health Survey, 2017-18PLOS ONE

Dear Dr. Zahura,

Thank you for submitting your manuscript to PLOS ONE. After careful consideration, we feel that it has merit but does not fully meet PLOS ONE’s publication criteria as it currently stands. Therefore, we invite you to submit a revised version of the manuscript that addresses the points raised during the review process.

One of the reviewers identified some minor concerns those need to be fixed before arriving final decision. 

We look forward to receiving your revised manuscript.

Kind regards,

Enamul Kabir

Academic Editor

PLOS ONE

Journal Requirements:

Reviewers' comments:

Reviewer's Responses to Questions

**Comments to the Author**

1. If the authors have adequately addressed your comments raised in a previous round of review and you feel that this manuscript is now acceptable for publication, you may indicate that here to bypass the “Comments to the Author” section, enter your conflict of interest statement in the “Confidential to Editor” section, and submit your "Accept" recommendation.

Reviewer #1: All comments have been addressed

Reviewer #2: All comments have been addressed

2. Is the manuscript technically sound, and do the data support the conclusions?

Reviewer #1: Yes

Reviewer #2: Yes

3. Has the statistical analysis been performed appropriately and rigorously? 

Reviewer #1: Yes

Reviewer #2: No

4. Have the authors made all data underlying the findings in their manuscript fully available?

Reviewer #1: Yes

Reviewer #2: Yes

5. Is the manuscript presented in an intelligible fashion and written in standard English?

Reviewer #1: Yes

Reviewer #2: No

6. Review Comments to the Author

Reviewer #1: All comments are addressed. Queries regarding methodology and conclusion has been corrected accordingly. Can procced for publication if other reviewers comments are addressed.

Reviewer #2: Comment #1: Abstract: Background: in the Abstract part well written and structured but in your paper abstract background part express a number of studies found the significant association of women’s 3E in maternal health outcomes, but no studies, to the best of knowledge, have been found to justify the joint influence of women’s 3E on the birth interval. As several studies have revealed that the short birth interval increases the risk of adverse maternal, perinatal and infant outcomes and it is also responsible for increasing the country’s population size, more research is needed on the birth interval. The above two paragraph is controversial and in your manuscript page four starting from paragraph four and the first paragraph of page five talks is several study including Bangladesh justify the joint influence of women’s 3E on the birth interval so, why you say there is no studies, to the best of knowledge, have been found to justify the joint influence of women’s 3E on the birth interval?

Comment #2: Abstract: Conclusion: In Conclusion part you recommended Policy-making interventions are needed to raise awareness among uneducated, under- empowered and economically poor reproductive women through family planning and fertility control programs so that the country can achieve the desired fertility rate, so say Conclusion and recommendation.

Comment #2: Method and Materials: Bangladesh Demographic and Health Survey (BDHS), 2017-18 data in Study design can you add reference of BDHS?

Comment #3: Method and Materials: table 1. Definition and measurements of the variables used in the study based on the 2017-18 BDHS: In the study, the education is converted into two categories: uneducated (no and primary), can we say primary education conclude as uneducated?

Comment #4: Method and Materials: what are your study participant’s inclusion and exclusion criteria?

Comment #5: Method and Materials: Statistical Analysis: what software uses to enter data and what software used to analysis data? Can you check goodness of fit tests? If you yes by what? You say Covariates having p-value less than 0.2 in the bivariate analysis were considered in the regression models. When you say statistically significant association independent and outcome variable? Please write detail in statistical analysis parts?

Comment #6: Method and Materials: Statistical Analysis: Explain how you went from OR to AOR, that is, what did you adjust for?

Comment #7: The English must be improved. There are numerous errors. Some words start with a capital letter for no reason. Spelling errors can be corrected using Word spell check.

Comment #9: Update references with the most recent

7. PLOS authors have the option to publish the peer review history of their article (what does this mean?). If published, this will include your full peer review and any attached files.

Reviewer #1: No

Reviewer #2: No

---

## [Author Response · Author response to Decision Letter 1]

26 Sep 2021

Response to reviewer 2

Comment 1: “Abstract: Background: in the Abstract part well written and structured but in your paper abstract background part express a number of studies found the significant association of women’s 3E in maternal health outcomes, but no studies, to the best of knowledge, have been found to justify the joint influence of women’s 3E on the birth interval. As several studies have revealed that the short birth interval increases the risk of adverse maternal, perinatal and infant outcomes and it is also responsible for increasing the country’s population size, more research is needed on the birth interval. The above two paragraph is controversial and in your manuscript page four starting from paragraph four and the first paragraph of page five talks is several study including Bangladesh justify the joint influence of women’s 3E on the birth interval so, why you say there is no studies, to the best of knowledge, have been found to justify the joint influence of women’s 3E on the birth interval?”

Response: Has been taken care of accordingly. Note that in literature, the variables women’s empowerment, education and economic status were used as covariates separately, whereas in this manuscript the joint effect of these three covariates was considered.

Comment 2: “Abstract: Conclusion: In Conclusion part you recommended Policy-making interventions are needed to raise awareness among uneducated, under- empowered and economically poor reproductive women through family planning and fertility control programs so that the country can achieve the desired fertility rate, so say Conclusion and recommendation.

”

Response: Has been adjusted accordingly.

Comment 2: “Method and Materials: Bangladesh Demographic and Health Survey (BDHS), 2017-18 data in Study design can you add reference of BDHS?”

Response: Reference was added.

Comment 3: “Method and Materials: table 1. Definition and measurements of the variables used in the study based on the 2017-18 BDHS: In the study, the education is converted into two categories: uneducated (no and primary), can we say primary education conclude as uneducated? 

Response: Topics related to maternal and child health care is not included in the curriculum of primary level of education in Bangladesh. Therefore, we considered primary education level in the uneducated group.

Comment 4: “Method and Materials: what are your study participant’s inclusion and exclusion criteria?”

Response: Inclusion and exclusion criteria have been provided in Figure 1.

Comment 5: “Method and Materials: Statistical Analysis: what software uses to enter data and what software used to analysis data? Can you check goodness of fit tests? If you yes by what? You say Covariates having p-value less than 0.2 in the bivariate analysis were considered in the regression models. When you say statistically significant association independent and outcome variable? Please write detail in statistical analysis parts?”

Response: The name of the Software has been added on Page 13. The goodness of fit test has also been performed (provided on Page 19) and result is included in Table 4. 

The reference has been included on Page 13 why p-value less than 0.2 was used. The p-value below 0.10 was used to define statistically significant association which was mentioned on Page 19.

Comment 6: “Method and Materials: Statistical Analysis: Explain how you went from OR to AOR, that is, what did you adjust for?”

Response: Note that in our manuscript, the term AOR was not used. The raised query in Comment 6 was addressed on Page 13 in Lines 223-224.

Comment 7: “The English must be improved. There are numerous errors. Some words start with a capital letter for no reason. Spelling errors can be corrected using Word spell check.”

Response: Has been taken care of. Microsoft word office 10 was used to prepare the draft.

Comment 9: “Update references with the most recent.”

Response: Has been updated.

---

## [Decision Letter · Decision Letter 2]

11 Jan 2022

Can Women’s 3E Index Impede Short Birth Interval? Evidence from Bangladesh Demographic and Health Survey, 2017-18

PONE-D-21-15599R2

Dear Dr. Zahura,

We’re pleased to inform you that your manuscript has been judged scientifically suitable for publication and will be formally accepted for publication once it meets all outstanding technical requirements.

Kind regards,

Enamul Kabir

Academic Editor

PLOS ONE

Additional Editor Comments (optional):

Reviewers' comments:

Reviewer's Responses to Questions

**Comments to the Author**

1. If the authors have adequately addressed your comments raised in a previous round of review and you feel that this manuscript is now acceptable for publication, you may indicate that here to bypass the “Comments to the Author” section, enter your conflict of interest statement in the “Confidential to Editor” section, and submit your "Accept" recommendation.

Reviewer #1: All comments have been addressed

Reviewer #2: All comments have been addressed

2. Is the manuscript technically sound, and do the data support the conclusions?

Reviewer #1: Yes

Reviewer #2: Yes

3. Has the statistical analysis been performed appropriately and rigorously? 

Reviewer #1: Yes

Reviewer #2: Yes

4. Have the authors made all data underlying the findings in their manuscript fully available?

Reviewer #1: Yes

Reviewer #2: Yes

5. Is the manuscript presented in an intelligible fashion and written in standard English?

Reviewer #1: Yes

Reviewer #2: Yes

6. Review Comments to the Author

Reviewer #1: All comments are addressed. Authors has corrected all the things according to the comment. Manuscript is written in suitable English

Reviewer #2: Thank you response for each point in the previous comments, now I see the manuscript detail, so this is a well-intentioned manuscript. Thanks with regards.

7. PLOS authors have the option to publish the peer review history of their article (what does this mean?). If published, this will include your full peer review and any attached files.

Reviewer #1: No

Reviewer #2: No

---

## [Editor Report · Acceptance letter]

17 Jan 2022

PONE-D-21-15599R2 

Can Women’s 3E Index Impede Short Birth Interval? Evidence from Bangladesh Demographic and Health Survey, 2017-18 

Dear Dr. Zahura:

I'm pleased to inform you that your manuscript has been deemed suitable for publication in PLOS ONE. Congratulations! Your manuscript is now with our production department. 

Kind regards, 

on behalf of

Dr. Enamul Kabir 

Academic Editor

PLOS ONE